

# Meter scale variation in shrub dominance and soil moisture structure Arctic arthropod communities

Rikke Reisner Hansen[1,2], Oskar Liset Pryds Hansen[1,2],
Joseph J. Bowden[1], Urs A. Treier[1,3,4], Signe Normand[1,3,4]
and Toke Høye[1,2,5]

[1] Arctic Research Centre, Department of Bioscience, Aarhus University, Aarhus C, Denmark
[2] Department of Bioscience, Kalø, Aarhus University, Rønde, Denmark
[3] Ecoinformatics & Biodiversity, Department of Bioscience, Aarhus University, Aarhus, Denmark
[4] Swiss Federal Institute for Forest, Snow and Landscape Research, Birmensdorf, Switzerland
[5] Aarhus Institute of Advanced Studies, Aarhus University, Aarhus, Denmark

## ABSTRACT

The Arctic is warming at twice the rate of the rest of the world. This impacts Arctic species both directly, through increased temperatures, and indirectly, through structural changes in their habitats. Species are expected to exhibit idiosyncratic responses to structural change, which calls for detailed investigations at the species and community level. Here, we investigate how arthropod assemblages of spiders and beetles respond to variation in habitat structure at small spatial scales. We sampled transitions in shrub dominance and soil moisture between three different habitats (fen, dwarf shrub heath, and tall shrub tundra) at three different sites along a fjord gradient in southwest Greenland, using yellow pitfall cups. We identified 2,547 individuals belonging to 47 species. We used species richness estimation, indicator species analysis and latent variable modeling to examine differences in arthropod community structure in response to habitat variation at local (within site) and regional scales (between sites). We estimated species responses to the environment by fitting species-specific generalized linear models with environmental covariates. Species assemblages were segregated at the habitat and site level. Each habitat hosted significant indicator species, and species richness and diversity were significantly lower in fen habitats. Assemblage patterns were significantly linked to changes in soil moisture and vegetation height, as well as geographic location. We show that meter-scale variation among habitats affects arthropod community structure, supporting the notion that the Arctic tundra is a heterogeneous environment. To gain sufficient insight into temporal biodiversity change, we require studies of species distributions detailing species habitat preferences.

Corresponding author
Rikke Reisner Hansen,
rrh@bios.au.dk

## INTRODUCTION

Understanding the factors that structure ecological communities on continental, regional and local scales provides the basis for understanding how global changes might affect

species composition and biodiversity (*Vellend et al., 2013*; *Dornelas et al., 2014*). Climate change is happening at an accelerated pace in the Arctic (*Callaghan et al., 2004*; *IPCC, 2014*), and altered moisture regimes and shrub expansion are two of the most prominent habitat-altering phenomena caused by these changes (*Rouse et al., 1997*; *Tape, Sturm & Racine, 2006*; *Myers-Smith et al., 2011*; *Elmendorf et al., 2012*). Shrub expansion and altered moisture regimes represent considerable consequences of climate change to the Arctic tundra; altering unique habitats such as open heath, wetlands and grasslands (*ACIA, 2004*). Firstly, warming in the Arctic has led to accelerated plant growth, particular for woody plants, causing a shift towards greater shrub cover, and a northward migration of the tree line (*Callaghan, Tweedie & Webber, 2011*), increased biomass (*Epstein et al., 2012*), and changes in plant species composition (*Walker et al., 2012*). These trends are expected to continue during future climate change (*Normand et al., 2013*; *Pearson et al., 2013*). Secondly, a changing Arctic climate with changes in precipitation, glacial melt, and permafrost degradation may alter the spatial extent of wetlands (*Avis, Weaver & Meissner, 2011*). In areas with continuous permafrost, top soils become wetter due to the impermeable strata that prevent infiltration and percolation (*Woo & Young, 2006*). Some areas with discontinuous permafrost, however, become drier, due to increased net evapotranspiration and increased drainage due to permafrost thaw (*Zona et al., 2009*; *Perreault et al., 2015*). The long term persistence of Arctic wetlands is debated, but climate change projections and field studies indicate deterioration and ultimate destruction of Arctic wetlands (*Woo & Young, 2006*). These habitat changes, both shrubification and wetland deterioration, will trigger several feedback loops within the climate system (*Chapin et al., 2005*) and may have profound effects on ecosystems (*Post et al., 2009*). In order to understand how these habitat changes, affect Arctic biodiversity, we need to adequately describe how Arctic species composition responds to environmental changes.

The alteration of habitats, due to e.g., shrub expansion into open tundra and changing wetland hydrology, are likely to affect habitat availability for many organisms, through changes in species' distributions, diversity, and composition. Terrestrial arthropods (e.g. insects and spiders) in particular, are associated with specific habitat types and likely respond strongly to habitat changes in the Arctic (*Bowden & Buddle, 2010*; *Rich, Gough & Boelman, 2013*). Arthropods have long been recognized as valuable indicators of changing environments because of their relatively short lifecycles and their physiology being directly driven by the external environment (ectothermic). Studies of the impacts of habitat changes upon Arctic arthropod communities are, however, only beginning to emerge (*Bowden & Buddle, 2010*; *Rich, Gough & Boelman, 2013*; *Sikes, Draney & Fleshman, 2013*; *Sweet et al., 2014*; *Hansen et al., 2016*). In spite of the common conception of the Arctic as a species-poor and relatively homogenous environment, studies have shown that arthropod assemblages vary substantially over short distances (*Hansen et al., 2016*), with species responding to local and regional climatic gradients (O. L. P. Hanser et al., 2013, unpublished data). Arthropod communities are expected to change in response to the direct effects of increasing temperatures and prolonged growing seasons (*Høye et al., 2013*; *Høye et al., 2014*), but also indirectly through changes in soil moisture and vegetation

structure (*Bowden & Buddle, 2010*; *Hansen et al., 2016*), changes to snowmelt dynamics (*Høye et al., 2009*; *Bowden et al., 2015b*), and shrub expansion (*Rich, Gough & Boelman, 2013*). Several studies indicate direct effects of temperature change on arthropods (*Post et al., 2009*; *Høye et al., 2013*; *Bowden et al., 2015a*), but we do not yet fully comprehend the distribution of, or habitat requirements for, the majority of Arctic arthropod species.

Arctic and alpine tundra areas are vast, and the knowledge of geographical variation associated with recent environmental and ecosystem change is limited. In this study, we explore the influence of moisture regime and habitat structure on the composition and diversity of Arctic arthropod communities, and investigate the site specific effects of the drivers of change. We propose the following hypothesis: Arctic arthropod assemblages and diversity vary with soil moisture and vegetation height at very small spatial scales (10–20 m). Specifically, we compare beetle and spider communities sampled in different habitats (fen, dwarf shrub heath, and tall shrub tundra) at three sites along a large scale gradient. We expect to find distinct arthropod communities in each habitat, and that abundances of groups like wolf spiders, and other active hunters, will be lower in the tall shrub tundra compared to open habitats.

## METHODS

### Study area and sampling design

Arthropods were sampled with uncovered pitfall traps from the 29th of June to the 23rd of July 2013 at three sites (1, 2, and 3) along the Godthaabsfjord in West Greenland (Fig. 1). Site 1 was situated at the mouth of the fjord and thus characterized by a coastal climate with relatively high precipitation, narrow annual temperature range, and topographic variation (app. 0–300 m.a.s.l.). The shrub community at site 1 was dominated by dwarf shrubs and a very sparse cover of tall shrub species such as *Salix glauca* (Lange (family: Salicidae)). Site 2 was low lying and flat, and characterized by a mosaic of low shrub vegetation (< 50 cm), dominated by *S. glauca*, mixed with *Betula nana* (Lange (family: Betulacae)), *Vaccinium uliginosum* (L., (family: Ericacae)), *Rhododendron groenlandicum* (Oeder (family: Ericacae)), and *Empetrum nigrum* (Lange (family: Ericacae)). Site 3 was characterized by a continental climate and pronounced topographic variation (app. 0–600 m.a.s.l.) with well-defined tall shrub patches dominated by high growth of *S. glauca* and *Alnus crispa* (Aiton (family: Betulacae)) (> 50 cm). These patches were mainly located at south facing slopes below 100 m.a.s.l. All dwarf shrub species at site 2 were also present at site 3.

Moisture transitions (fen-heath) were sampled at sites 1 and 2, while transitions in vegetation height and cover of tall shrubs (heath-shrub) were sampled at sites 2 and 3. Four fen-heath plots were established, two at site 1 and two at site 2. Each fen-heath plot consisted of two sub-plots placed 10 m apart. Each sub-plot was situated exactly 5 m from a distinct fen-heath transition zone (Fig. 2). Twelve heath-shrub plots were established at site 2 and site 3 (six at each site). Each heath-shrub plot consisted of two sub-plots 20 m apart; one located at the center of a patch of tall shrubs and one in the adjacent open dwarf shrub heath. Each sub-plot was delineated by a circle with a 5 m radius. At the center of each sub-plot, two yellow pitfall traps (9 cm diameter) were placed 50 cm

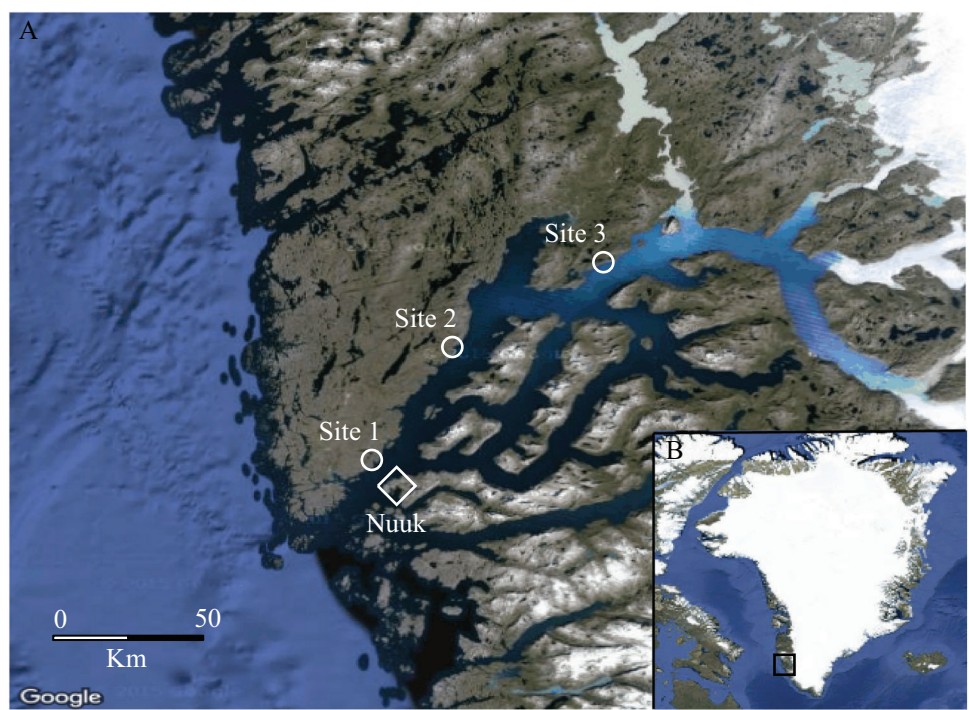

**Figure 1 Map of the study area.** (A) The Godthåbsfjord area, South-West Greenland (64°11′N, 51°44′W), showing the three study sites (1, 2 and 3) depicted with a circle and the capital Nuuk depicted with a diamond. (B) Greenland with the study area framed in a square (*Loecher & Ropkins, 2015*). Mapdata: © Google 2016.

apart (Fig. 2). The traps were dug down such that the rim was flush with the surface and filled one third with a soap water solution. There was no overflow due to rainwater accumulation during sampling. The color of the pitfalls was chosen to catch flying as well as surface-active arthropods (*Høye et al., 2014*). Pitfall traps were emptied twice, once halfway through and once at the end of the sampling period. Samples were stored separately.

The following structural and environmental parameters were measured in each sub-plot: (i) percent cover of shrubs, herbs, graminoids and bare ground in six categories: 0, 1–20, 21–40, 41–60, 61–80, and 81–100%, (ii) height (to the nearest 5 cm) of the vegetation height with the highest coverage in the sub-plot, (iii) presence of plant species, (iv) slope in vertical meters between the highest and lowest point of the sub-plot, (v) aspect, recorded using a handheld GPS and classified to nearest cardinal direction (North, South, East, and West), (vi) pH, measured directly with a soil pH measurement kit, model HI 99121, (vii) soil type at 15 cm depth was recorded as humus or sand.

## Specimens and data

All spiders and beetles were sorted from the samples and the adult specimens were identified (by RRH) to species based on morphological characters using a Wild® M5A stereo microscope. Not all juveniles could be assigned to species, so only adult specimens were included in the analysis. Spiders were identified using the available literature through The *World Spider Catalog (2016)* and Spiders of North America (*Paquin & Dupérré, 2003*).

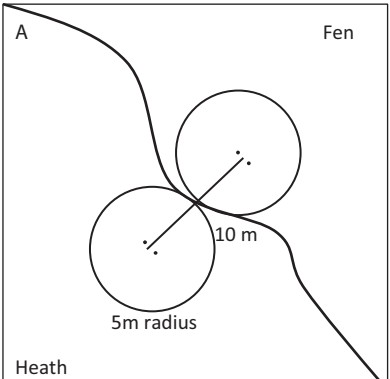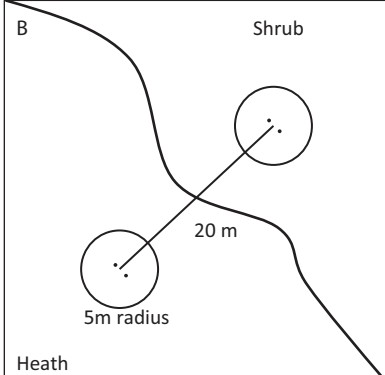

**Figure 2 Sampling design.** Conceptual figure of the sampling design showing fen transects in the (A) and shrub transects in the (B).

Beetles were identified using both Scandinavian and North American literature (*Lindroth, 1985*; *Lindroth, 1986*; *Böcher, 1988*) and by consulting the collection at the Natural History Museum Aarhus, Denmark. Specimens are preserved in 75% ethanol at the Natural History Museum Aarhus. The dataset is available through the Global Biodiversity Information Facility (http://doi.org/10.15468/li6jkm).

## Data analysis

The mean and standard error were calculated for significant environmental variables across all habitats at each site. We ran a correlation analysis of all potential variables, based on Pearson's correlation coefficient, and tested whether the uncorrelated variables differed significantly between sites and habitats with a MANOVA. To counteract effects of potential-under sampling, all analyses were carried out excluding singletons. All analyses were carried out in R version 3.2.2.

## Species diversity

Species diversity was rarefied and extrapolated for investigation across habitats based on Hill numbers ($q = 0$; species richness, $q = 1$; Shannon diversity, $q = 2$; Simpson diversity) and standardized by sample coverage (*Chao & Jost, 2012*; *Chao et al., 2014*) using the iNEXT R-package (TC Hsieh, KH Ma & A Chao, 2014. unpublished data). We extrapolated to double the reference sample of the habitat with the smallest sample coverage (shrub). Samples were compared at base-coverage, estimated as a minimum of $C_a$ and $C_b$, where $C_a$ is maximum coverage at reference sample size and $C_b$ is minimum coverage at two times reference sample size. iNEXT computes bootstrap confidence bands around the sampling curves, facilitating the comparisons of diversity across multiple assemblages. We then visually assessed if diversity measures differed significantly between habitats.

We ran a species indicator analysis to assess the strength and statistical significance of the relationship between species abundances and the specific habitats. We used the function 'multipatt' in the R package 'indicspecies' (*De Cáceres, Legendre & Moretti, 2010*). This analysis provides a specificity value 'A'(0–1), which indicates the probability of a certain species occurring in a certain habitat as well as a sensitivity value 'B'(0–1), which

indicates how many of the plots belonging to a certain habitat the target species is located in. Significance ($p < 0.05$) is assessed based on the A and B values (*De Cáceres & Legendre, 2009*). In addition to significance testing, we opted to describe habitat preferences more broadly by assigning all species with an A value for a given habitat larger than 0.8 and a B value larger than 0.1 to that specific habitat. In this way, the importance of the sensitivity value is down weighted.

## Species composition

Traditional methods to visually investigate how arthropod species composition varies between habitats, such as non-metric multidimensional scaling (NMDS), have been shown to confound trends in location with changes in dispersion, leading to potentially misleading results (*Warton, Wright & Wang, 2012*). To avoid these issues while still enabling visualization, we employed latent variable modelling through the R package 'boral' (*Hui, 2016*). Latent variable modelling is a Bayesian model-based approach that explains community composition through a set of underlying latent variables to account for residual correlation, for example due to biotic interaction. This method offers the possibility to adjust the distribution family to e.g., negative binomial distribution which better accounts for over-dispersion in count data. Thus, it accounts for the increasing mean-variance relationship without confounding location with dispersion (*Hui et al., 2015*). Three "types" of models may be fitted: (1) With covariates and no latent variables, boral fits independent response General Linear Models (GLMs) such that the columns of y are assumed to be independent; (2) With no covariates, boral fits a pure latent variable model (*Rabe-Hesketh, Skrondal & Pickles, 2004*) to perform model-based unconstrained ordination (*Hui et al., 2015*); (3) With covariates and latent variables, boral fits correlated response GLMs, with latent variables.

Sub-plots placed in dwarf shrub heath could potentially differ depending on the transition examined (fen-heath or shrub-heath). Therefore, we created latent variable plots for both plants and arthropods to visually assess if the heath sub-plots in the fen-heath and shrub-heath plots groups were distinguishable. In the latent variable plot for plant species composition, heath sub-plots were not segregated (Fig. S1) and all heath plots were hereafter treated as one category.

We modelled arthropod species' distributions with two latent variables to enable visualization comparable to a two dimensional NMDS. From the latent variable model, we extracted the posterior median values of the latent variables which we used as coordinates on ordination axes to represent species composition at plot level (*Hui et al., 2015*). We then tested the difference in local species composition based on these coordinates between the paired samples (fen-heath or shrub-heath) for each transect using paired t-tests.

To test the significance of and, interactions between, the environmental variables and site, we used a multivariate extension of GLMs using the function 'manyglm' in the package 'mvabund' (*Wang et al., 2012*). This recently developed method offers the possibility to model distributions based on count data by assuming a negative binomial distribution. Vegetation height and graminoid cover have higher resolutions compared to

the classifications 'fen' and 'shrub' as these are measured on a continuous scale. We used vegetation height as a proxy for shrub treatment effects and cover of graminoids as a proxy for soil moisture. The gradients in these variables are representative of the moisture transition of fen-heath plot groups and the shrub dominance transition of the shrub-heath plot groups (Fig. S2). We tested for main effects of all the un-correlated variables, selected by the Pearson correlation analyses, and for an interaction between the variables and site. The model assumptions of mean-variance and log-linearity were examined with residual vs. fit plots and a normal quantile plot, and no transformations were needed.

## RESULTS

A total of 2,547 individuals, constituting 45 species and 13 families were identified within the two orders: Araneae (2,223 individuals, seven families, 37 species) and Coleoptera (324 individuals, 6 families, 8 species). We found a species of sheet web spider (*Wabasso cacuminatus* (Millidge, 1984)) not previously known from Greenland, represented by one individual. One species (*Pelecopsis mengei*, (Simon, 1884)), represented in our samples by three individuals, remained unknown from Greenland until recently (*Marusik, 2015*; *Hansen et al., 2016*) (Table 1).

Extrapolated species richness ($q = 0$) did not differ significantly between habitats due to overlapping confidence intervals but there was a trend towards higher species richness in heath sub-plots, lower in shrub sub-plots, and lowest in fen sub-plots (Fig. 3). The same pattern was observed for Shannon diversity ($q = 1$) as well as for Simpson diversity ($q = 2$), however both these indices differed significantly between habitats, with the highest diversity in the shrub sub-plots, intermediate in the heath sub-plots, and lowest in the fen sub-plots (Fig. 3).

The three species significantly ($p < 0.05$) associated with fen habitats were all sheet web spiders. *Erigone whymperi* (O.P. Cambridge, 1877), *Mecynargus paetulus* (O.P. Cambridge, 1875), and *Wabasso quaestio* (Chamberlin, 1948). Just one species, the ladybird *Coccinella transversoguttata* (Faldermann, 1835), was significantly associated with heath habitats. Shrub habitats housed six significantly associated species: the comb-footed spider *Ohlertidion lundbecki* (Sørensen, 1894), and five species of sheet web spiders: *Dismodicus decemoculatus* (Emerton, 1852), *Improphantes complicates* (Emerton, 1882), *Pocadicnemis americana* (Millidge, 1984), *Semljicola obtusus* (Emerton, 1914), *Sisicus apertus* (Holm, 1939) (Table 1).

The latent variable plots showed that the plant species composition of the shrub sub-plots overlapped with the composition of the heath plots (Fig. S1), but vegetation height was significantly different (Table 2). The plant species composition of the fen plots was different from both the heath and shrub sub-plots (Fig. S1). Arthropod species composition was segregated both at site and habitat level, but the distribution of sub-plots in the latent variable arthropod plot indicated interaction between site and treatment (Fig. 4).

Vegetation height in the shrub sub-plots at site 2 was significantly lower than that at site 3 ($F_{1,21} = 13.46$, $p = 0.001$), and the overall cover of graminoids was significantly lower at the fen sub-plots at site 1, compared to the fen sub-plots at site 2 ($F_{1,29} = 0.21$,

Table 1 **Arthropod species.** List of arthropod species sampled and their abundance in three habitats; fen, dwarf shrub heath, and tall shrub tundra at three sites along the Nuuk fiord in Western Greenland. The last column shows the results of a species indicator analysis (for details see main text). Species were assigned to one of the three habitats if A (specificity value) > 0.8 and B (sensitivity value) > 0.1. The table is sorted by order, family, and species, respectively.

| Order | Family | Species | Abundance | | | Habitat |
|---|---|---|---|---|---|---|
| | | | Fen | Heath | Shrub | |
| Araneae | Dictynidae | *Dictyna major* (Menge, 1869) | | | 1 | No classification |
| | Gnaphosidae | *Haplodrassus signifer* (C.L. Koch, 1839) | | 1 | | No classification |
| | Hahniidae | *Hahnia glacialis* (Sørensen, 1898) | 1 | 7 | 1 | No classification |
| | Linyphiidae | *Agyneta jacksoni* (Simon, 1884) | 3 | 8 | 1 | No classification |
| | | *Agyneta nigripes* (Brændegård, 1937) | 2 | 3 | | Fen and heath |
| | | *Bathyphantes simillimus* (L. Koch, 1879) | | | 1 | No classification |
| | | *Dismodicus decemoculatus* (Emerton, 1852) | 1 | 2 | 10 | Shrub* |
| | | *Erigone arctica* (White, 1852) | 6 | | | Fen |
| | | *Erigone psycrophila* (Thorell, 1871) | 1 | | | No classification |
| | | *Erigone whymperi* (O.P. Cambridge, 1877) | 8 | | | Fen* |
| | | *Hilaira herniosa* (Thorell, 1875) | | 1 | | No classification |
| | | *Hybauchenidium gibbosum* (Sørensen, 1898) | | 5 | 3 | Heath and shrub |
| | | *Hypsosinga groenlandica* (Simon, 1889) | 2 | 2 | 4 | Heath and shrub |
| | | *Improphantes complicatus* (Emerton, 1882) | | 2 | 8 | Shrub* |
| | | *Lepthyphantes turbatrix* (O.P. Cambridge, 1877) | | | 1 | No classification |
| | | *Mecynargus borealis* (Jackson, 1930) | | 4 | | Heath |
| | | *Mecynargus morulus* (O.P. Cambridge, 1873) | | 2 | 1 | Heath and shrub |
| | | *Mecynargus paetulus* (O.P. Cambridge, 1875) | 33 | | | Fen* |
| | | *Oreonetides vaginatus* (Thorell, 1872) | | | 1 | No classification |
| | | *Pelecopsis mengei* (Simon, 1884) | | 2 | 1 | Heath and shrub |
| | | *Pocadicnemis americana* (Millidge, 1976) | | 6 | 18 | Shrub* |
| | | *Sciastes extremus* (Holm, 1967) | | 1 | | No classification |
| | | *Scotinotylus sacer* (Crosby, 1929) | | | 5 | Shrub |
| | | *Semljicola obtusus* (Emerton, 1914) | 3 | 6 | 15 | Shrub* |
| | | *Sisicus apertus* (Holm, 1939) | | 1 | 3 | Shrub* |
| | | *Tiso aestivus* (L. Koch, 1872) | 1 | 31 | 1 | Heath |
| | | *Wabasso cacuminatus* (Millidge, 1984) | | 1 | | No classification |
| | | *Wabasso quaestio* (Chamberlin, 1948) | 12 | | | Fen* |
| | | *Walckenaeria karpinskii* (O.P. Cambridge, 1873) | 6 | 21 | | Fen and heath* |
| | Thomisidae | *Xysticus durus* (Sørensen, 1898) | | 17 | | Heath |
| | Lycosidae | *Arctosa insignita* (Thorell, 1872) | 17 | 29 | 2 | Fen and heath* |
| | | *Pardosa furcifera* (Thorell, 1875) | 524 | 552 | 257 | No classification |
| | | *Pardosa groenlandica* (Thorell, 1872) | 17 | 23 | 8 | No classification |
| | | *Pardosa hyperborea* (Thorell, 1872) | 6 | 347 | 140 | Heath and shrub* |
| | Philodromidae | *Thanatus arcticus* (Thorell, 1872) | 2 | 10 | | Fen and heath |
| | Theridiidae | *Robertus fuscus* (Emerton, 1894) | | | 1 | No classification |
| | | *Ohlertidion lundbecki* (Sørensen, 1898) | | | 2 | Shrub |

| Order | Family | Species | Abundance | | | Habitat |
|---|---|---|---|---|---|---|
| | | | Fen | Heath | Shrub | |
| Coleoptera | Byrrhidae | *Byrrhus fasciatus* (Forster, 1771) | 1 | 11 | | Heath |
| | Carabidae | *Patrobus septentrionis* (Dejean, 1821) | 50 | 17 | 23 | Fen and shrub* |
| | Coccinellidae | *Coccinella transversoguttata* (Falderman, 1835) | | 51 | 2 | Heath* |
| | Cryptophagidae | *Caenoscelis ferruginea* (Sahlberg, 1820) | | 38 | 2 | Heath and shrub |
| | Curculionidae | *Otiorynchus arcticus* (O. Fabricius, 1780) | 1 | 20 | 1 | Heath |
| | | *Otiorynchus nodosus* (Müller, 1764) | 18 | 66 | 19 | No classification |
| | Staphylinidae | *Mycetoporus nigrans* (Mäklin, 1853) | | 2 | | No classification |
| | | *Quedius fellmanni* (Zetterstedt, 1838) | | 2 | | No classification |

**Note:**
*Indicates Significance, $p < 0.05$.

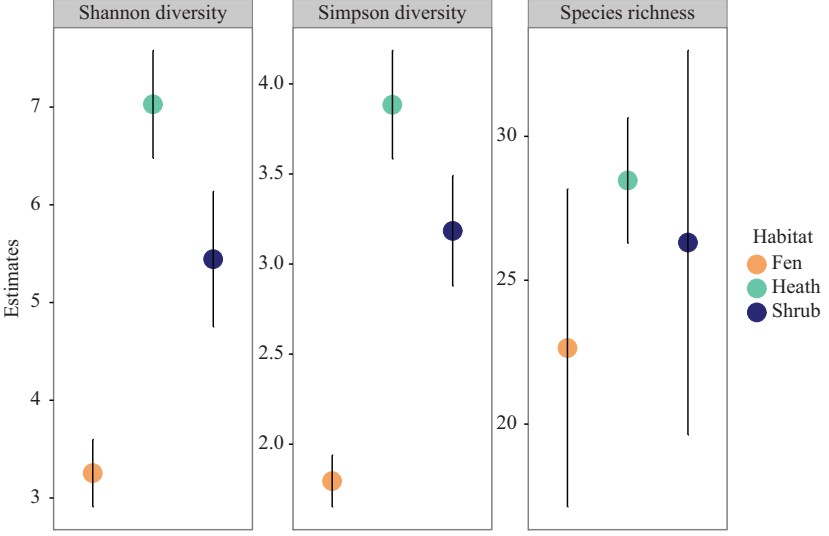

**Figure 3 Diversity profiles.** Species richness, Shannon diversity and Simpson diversity coloured by habitat. Error bars represent 95 percent confidence intervals.

**Table 2 Environmental variables.** Mean (± S.E) of the environmental variables included in GLM's and latent variable models, showing the difference between sites and treatments. Graminoid cover was measured in six categories: 0, 1–20, 21–40, 41–60, 61–80, and 81–100%. Vegetation height was measured (classified to the nearest 5 cm) of the vegetation height with the highest coverage in the sub-plot.

| Site | Habitat | Vegetation height (height classes) | Graminoid (% cover) | pH |
|---|---|---|---|---|
| Site 1 | Heath | 2.6 (0.2) | 15 (5) | 5.8 (0.1) |
| | Fen | 2.5 (0.2) | 55 (6.3) | 5.5 (0.2) |
| Site 2 | Heath | 2.4 (0.2) | 18.6 (3.7) | 6.4 (0.1) |
| | Fen | 2.3 (0.3) | 75 (6.3) | 6.5 (0.04) |
| | Shrub | 7.5 (1.2) | 10.3 (3.5) | 6.2 (0.3) |
| Site 3 | Heath | 3.2 (0.4) | 12.7 (11.4) | 6.2 (0.2) |
| | Shrub | 28.5 (4.1) | 4 (1.9) | 6.5 (0.04) |

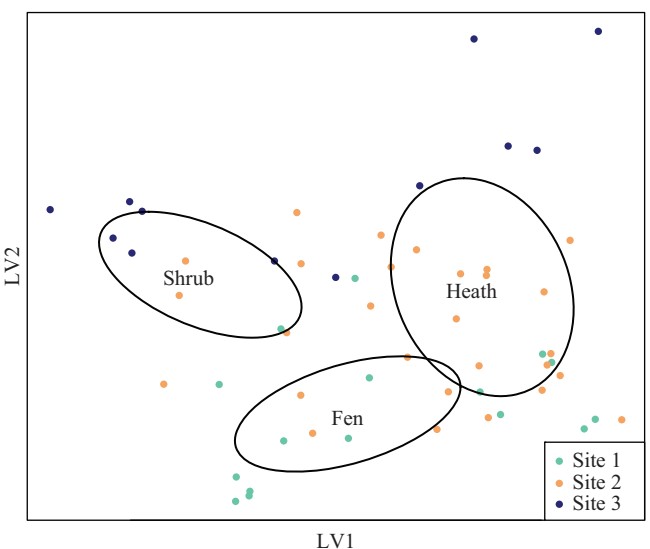

**Figure 4 Latent variable plot for arthropod species composition.** Species distribution plot of the best fitted latent variable model showing the mean of the latent variable with a negative binomial distribution. Ellipses represent 95 percent confidence intervals around the centroids of each habitat.

**Table 3 Table of deviance.** Results of the multivariate generalized linear model, including all variables tested, along with residual degrees of freedom, degrees of freedom, deviance and *p*-value.

| Parameter | Residual degrees of freedom | Degrees of freedom | Deviance | *p*-value |
|---|---|---|---|---|
| Intercept | 55 | | | |
| Vegetation height (height class) | 54 | 1 | 117.9 | 0.001 |
| Graminoid cover (% cover) | 53 | 1 | 93.2 | 0.001 |
| pH | 52 | 1 | 43.0 | 0.389 |
| Site | 50 | 2 | 296.2 | 0.001 |
| Vegetation height: site | 48 | 2 | 35.0 | 0.639 |
| Graminoid cover: site | 46 | 2 | 103.8 | 0.003 |
| pH: site | 44 | 2 | 40.8 | 0.568 |

$p = 0.049$). The pH levels were nearly significantly different between the shrub sub-plots from site 2 to site 3 ($F_{1,21} = 4.17$, $p = 0.054$) and highly significantly different between the fen sub-plots from site 1 to site 2 ($F_{1,29} = 66.01$, $p < 0.0001$). pH did not differ between fen and heath ($F_{1,29} = 0.69$, $p = 0.41$) or shrub and heath ($F_{1,21} = 0.50$, $p = 0.49$). Cover of graminoids was significantly lower for heath sub-plots compared to fen sub-plots ($F_{1,29} = 74.54$, $p < 0.0001$). Vegetation height differed significantly between shrub and heath habitats ($F_{1,21} = 26.04$, $p < 0.0001$), with lower vegetation height in the heath sub-plots compared to shrub sub-plots (Table 2; Fig. S2).

Arthropod species composition differed significantly due to different moisture regimes and different height classes. pH levels were not a significant driver of arthropod communities, nor was there a significant interaction between site and the levels of pH.

**Table 4 T-test of local transitions.** Paired *t*-test of the local transitions in soil moisture and shrub dominance. LV1 and LV2 represent the first and second coordinate of the latent variable.

| Model | Residual degrees of freedom | Estimates | t | p |
|---|---|---|---|---|
| Fen transect site 1 LV1 | 7 | −0.86 | −5.32 | 0.001 |
| Fen transect site 1 LV2 | 7 | −0.43 | −4.78 | 0.002 |
| Fen transect site 2 LV1 | 7 | −1.70 | −0.26 | 0.13 |
| Fen transect site 2 LV2 | 7 | −0.37 | −3.21 | 0.02 |
| Shrub transect site 2 LV1 | 5 | −0.72 | −3.90 | 0.01 |
| Shrub transect site 2 LV2 | 5 | −0.35 | −3.10 | 0.03 |
| Shrub transect site 3 LV1 | 5 | −1.16 | −5.50 | 0.003 |
| Shrub transect site 3 LV2 | 5 | −0.62 | −3.28 | 0.02 |

There was a significant interaction between cover of graminoid species and site, but no significant interaction was detected between height class and site (Table 3). Arthropod species composition differed significantly between the local fen-heath transitions, but for site 2 only; one latent variable axis differed significantly between fen-heath transitions. The local shrub-heath transects differed significantly for both axes and both sites (Table 4).

## DISCUSSION

Although Arctic tundra is often perceived as a relative homogenous biome, it consists of a wide range of habitat types due to strong environmental transitions occurring over short spatial scales. In this study, we have demonstrated clear effects of vegetation height and soil moisture on diversity and composition of spiders and beetles in low Arctic Greenland. This effect is evident across distances of 10–20 m. Fen, heath, and shrub vegetation hosted distinct arthropod communities differing in both composition and diversity. While previous studies have emphasized the importance of vegetation structure as predictors of Arctic arthropod communities (*Bowden & Buddle, 2010*; *Rich, Gough & Boelman, 2013*; *Sweet et al., 2014*), it has not been demonstrated that such effects are visible at the scale of meters.

Existing literature generally agrees with the habitat classifications we have assigned the species in this study. According to existing descriptions of habitat preferences, the wetland species we find in this study are found strictly in wet open habitats, whereas both shrub and heathland species mostly have a more general distribution (*Leech, University of Alberta & Department of Entomology, 1966*; *Böcher, 2015*; *Marusik, 2015*), indicating a higher degree of habitat specialization in the fens. The sheet web spider, *Erigone arctica*, was significantly linked to wet fen habitats in an alpine study site in West Greenland (*Hansen et al., 2016*), and in this study, *E. arctica* was also linked to fen plots, further suggesting habitat specialization. We found the lowest diversity in the fens, which are spatially limited, compared to much more wide spread heathland habitats. Both tall shrub tundra and dwarf shrub heath are comprised of different habitats with open patches, moist areas, and varying vegetation structure. Such variation in habitat structure likely leads to higher diversity compared to the fen habitats, which are rather homogenous.

This particular study area is characterized as low Arctic with discontinuous permafrost unaffected by glacial meltwater. Models predict that this area will experience increased evapotranspiration and precipitation (*Rawlins et al., 2010*). Increased drainage due to permafrost melt coupled with evapotranspiration is likely to lead to wetland deterioration. Shrubification has been forecasted to be most pronounced at the boundary between high and low Arctic where permafrost is melting and in areas where soil moisture is greatest (*Myers-Smith et al., 2015*). In the Godthåbsfjord, it is therefore likely that shrub expansion will be most notable in the fens and snow-beds. With shrubification (*Myers-Smith et al., 2011*; *Elmendorf et al., 2012*), as well as, increased land use such as forestry and agriculture (*ACIA, 2004*), wetland habitats are at risk (*CAFF, 2013*). Our results suggest that wetland deterioration and shrubification will strongly affect arthropod  communities, and may compromise the living conditions of individual specialized species.

We found an interaction between site and graminoid cover, suggesting that the fens differ between sites. Wetlands with coastal proximity are known to be impacted by salt influx from the sea (*Woo & Young, 2006*). This is supported by the slightly higher pH at site 1 compared to site 2, but does not explain differences in arthropod composition in the fens between the coastal (site 1) and intermediate site (site 2), as pH was not significant in the multivariate GLM. Even though plant species composition showed clear segregation of wet and dry plots, conditions may be drier at the intermediate site than at the coastal site, where summer precipitation is higher. Plant species composition reflects an integration of seasonal variation in soil moisture conditions (*Daniels et al., 2011*) such that they may not reflect sudden soil moisture changes. The variation in moisture regime only partially explained arthropod species composition at the intermediate site and supports the idea of drier conditions at the intermediate site affecting arthropod species composition differently.

We expected the effect of vegetation height to be less pronounced at the intermediate site due to the patchy structure of the shrubs and overall lower vegetation; yet, we did not find an interaction between site and treatment. We studied mostly mobile predatory species. The few herbivores like the weevils: *Otiorynchus arcticus* (O. Fabricius, 1780) and *Otiorynchus nodosus* (Müller, 1764) are mostly found in open heath plots with low vegetation. It is conceivable that even a small change in vegetation height has an effect on the surface active predatory species, because vegetation height may also affect the composition and abundance of prey items. The web builders, like sheet web spiders, require some amount of vegetation structure to form webs, but even low shrubs provide structure and shelter. *Rich, Gough & Boelman (2013)* found that overall arthropod abundance and species richness increased in shrub plots in Arctic Alaska, but suggested that for groups like wolf spiders and other active hunters, full shrub encroachment of open habitats could be detrimental. Our results show that abundances of these groups are lower in shrub sub-plots and support this notion.

## CONCLUSION

We have established a baseline of species occurrence in relation to transition in soil moisture and shrub dominance which will facilitate future assessment of changes in Arctic

arthropod communities, where these transitions in habitat structure are likely to change. The variation in community composition at the scale of meters was surprising and suggests drastic changes in arthropod species composition given continued shrubification and wetland deterioration. We found that the strength of the environmental predictor variables varied among sites. Understanding the sources of such site variation is an important topic for future studies. Two important steps are needed to further the knowledge of arthropod responses to changing habitats. Primarily, we need information on species occurrence across multiple taxa and multiple environmental gradients. Habitat preferences of species are needed to determine the effects that climate change will have in Arctic ecosystems. Spiders and butterflies have proven useful for detection of rapid environmental change due to climate change (*Høye et al., 2014*; *Bowden et al., 2015a*; *Bowden et al., 2015b*) and may serve as important indicator taxa in future studies. Secondly, we need further studies of spatial variability and change in environmental gradients like soil moisture.

## ACKNOWLEDGEMENTS

This work is a contribution to the Arctic Science Partnership (ASP: http://asp-net.org). We would also like to thank the Natural History Museum Aarhus for use of laboratory and equipment during the identification process.

### Funding

Toke Thomas Høye was supported by a Jens Christian Skou fellowship at the Aarhus Institute of Advanced Studies and Signe Normand was supported by the Villum foundation's Young Investigator Programme (VKR023456). Logistical support was provided by the Arctic Research Centre (ARC), Aarhus University. The funders had no role in study design, data collection and analysis, decision to publish, or preparation of the manuscript.

### Grant Disclosures

The following grant information was disclosed by the authors:
Jens Christian Skou Fellowship at the Aarhus Institute of Advanced Studies.
Villum foundation's Young Investigator Programme: VKR023456.
Arctic Research Centre (ARC), Aarhus University.

### Competing Interests

The authors declare there are no competing interests.

### Author Contributions

- Rikke Reisner Hansen conceived and designed the experiments, performed the experiments, analyzed the data, wrote the paper, prepared figures and/or tables, reviewed drafts of the paper.

- Oskar Liset Pryds Hansen performed the experiments, contributed reagents/materials/ analysis tools, reviewed drafts of the paper, contributed substantially to the design of the experiments, contributed to data interpretation and approved the final version.
- Joseph J. Bowden contributed reagents/materials/analysis tools, reviewed drafts of the paper, contributed substantially to data interpretation and approved the final version.
- Urs A. Treier reviewed drafts of the paper, contributed to the fieldwork, data interpretation and approved the final version.
- Signe Normand reviewed drafts of the paper, contributed to the fieldwork, data interpretation and approved the final version.
- Toke Høye conceived and designed the experiments, reviewed drafts of the paper, contributed substantially to the fieldwork, data interpretation and approved the final version.

### Data Deposition

Global Biodiversity Information Facility:

http://www.gbif.org/dataset/59d25ddf-355d-47d7-b8a1-c1e2819014c7

DOI: 10.15468/li6jkm

### Supplemental Information

Supplemental information for this article can be found online at http://dx.doi.org/10.7717/peerj.2224#supplemental-information.

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
