# Peer review of "Meter scale variation in shrub dominance and soil moisture structure Arctic arthropod communities"

_PeerJ, doi:10.7717/peerj.2224_

## Round 0.1 · original submission · Minor Revisions

Thank you for submitting this interesting MS to PeerJ. From the two reviews and my own reading, I am assigning this to minor revisions. While there are some very important comments expressed about the analyses, I believe that most or all of them can be addressed by better explanation of the statistical methods and perhaps some minor reanalyses. All of the necessary data seem to be present (i.e., no one is asking for any further data collection) to allow this to happen with minimal effort. However, it will be vital for the authors to rigorously defend their analytical decisions in their rebuttal letter accompanying their revised submission.

I do have a few other comments, some of which echo the excellent reviews provided by our two reviewers (thanks to both of you for your excellent and detailed work). I have also attached a marked-up PDF. I used the already-marked PDF supplied in Dr. Buckley's review and added my own comments and edits. So the PDF attached to my report should show all of the suggestions from Dr. Buckley and myself in one file.

In addition:

Please use Oxford commas throughout. I have tried to note all locations where these should be added. In some cases the authors did this, but in many they did not. There should at least be consistency in this regard, and I'd suggest consistent use of Oxford commas to reduce misreading of the text in certain instances.

GPS locations of plots (or at least sites… but I’d recommend at the plot level) would be useful and should be added either in the text or – perhaps preferably – in a table. The map is decent, but since the authors have the precise data, a table with the exact locations would add value, particularly if someone else wants to come back to the exact same locations for temporal comparative work in the future.

Pitfall traps were not covered with a canopy or other device? What was the fluid (if any) in the cups? I assume rainwater accumulation if not covered. Was fluid accumulation less than that required to overflow the cups? A slight bit more detail on the traps would be useful to better understand the trapping effort.

As mentioned by Dr. Wheeler, the DOI for GBIF dataset is required.

Line 117: Do you mean “…with each subplot at least five meters from a distinct fen-heath transition zone…” Or do you mean exactly five meters in nearest measurement? Or are you talking about the plots (not the sub-plots)?

Line 173: Do you mean "...assign habitat categories to more species..." ?

Line 232 to 236, suggested wording: “Shrub habitats housed six significantly associated species, and all of them were spiders. They were the comb-footed spider Ohlertidion lundbecki (Sørensen, 1894), and five species of sheet web spiders: Dismodicus decemoculatus, Improphantes complicates, Pocadicnemis americana, Semljicola obtusus, and Sisicus apertus (Table 1).”

Thanks once again for this submission, and thank you to the reviewers for their excellent work. I look forward to your revised version and rebuttal to all of the comments and suggestions provided by the reviewers.

·

Basic reporting

This work should be reviewed for word usage, spelling, grammar and clarity. The background and introduction are sufficient. The figures are clear except figure three, which should be plotted as three facets where the y-axis scales are free to vary because the variables are not on the same scale.

Experimental design

The aims of the paper and the experimental design are appropriate. The data collection was conducted in a rigorous way. I have some difficulties understanding some aspects of the statistical analysis methods, which need to be clarified. Similarly, I there are some aspects of the results that aren't clear to me, mostly because I couldn't understand what had been done. See my detailed comments below and on the pdf.

Validity of the findings

The results seem to be valid, but some of the analyses are hard to judge because I found it hard to understand what was done in some cases (see comments below).

Additional comments

This paper is a useful contribution to the literature in that it provides a baseline for invertebrate composition studies in the future in these dynamic habitats. There were some aspects of the methods and presentation of the results that I found confusing.

1. Some of the statistical method descriptions are unclear:
a. Explain the latent variable method in full. Readers want to know what the method does and how it works.
b. Why was a t-test used rather than a two-way ANOVA to compare sites and habitats?
c. Explain the LV modelling in full (see detailed comments on the pdf file for more details).
d. Using significance testing and AIC model selection approaches in the same paper is incongruous. Why switch between correlations and GLMs with AIC model selection? Is it necessary to present both approaches?
e. If you used ‘site’ as a ‘blocking effect’ then did you use generalised linear mixed models?
2. Diversity results should be explained in full so readers know what the differences were (line 228-229)
3. Can you colour the map? It’s hard to tell what is land and what is water.
4. Plot figure three as three separate graphs in a facet where the y axis is free. The scales of richness, Shannon’s and Simpsons are not comparable.
5. Fig. 5: I don’t understand what this plot represents. What does ‘plot of the correlations due to environmental responses mean’? Normally red is positive and blue is negative. I would swap these colours.
6. Why do your habitat types differ in between some graphs and the supplementary figure (dry, fen, open, shrub vs. shrub, fen, heath)?
7. The manuscript should be edited for word usage and clarity (see comments on the pdf).
8. Results: I would avoid calling the habitat types ‘treatments’ since this was an observational study. The use of this term throughout the paper is confusing.
9. What do you mean by ‘residual species correlations’?
10. I did not get a strong impression of what the important environmental drivers in the system were. Is it possible to include a figure or table that describes how the LV results relate to the environmental variables you measured more clearly that what you already have? Your discussion mentions salinity and other soil factors. It would be nice to see how important the abiotic variables (slope, aspect, pH, moisture) you measured were to composition. A constrained ordination approach would be useful for this. I can’t understand how the LV analysis is useful for understanding the species-environment relationships.

Please see my detailed comments on the attached pdf.

·

Basic reporting

Ongoing ecological research in the arctic is dismantling the myth that arctic ecosystems are homogeneous habitats with low biodiversity and simple food webs. This manuscript addresses the complexity of arctic arthropod assemblages at small spatial scales in Greenland and demonstrates that diversity in the far north is not as simple as “conventional wisdom” assumes.

The manuscript is clearly written and concise. The Background section provides good justification for the necessity of baseline data, at small spatial scales, on the structure of arctic arthropod communities in the face of ongoing and significant climate change impacts. The literature cited is relevant and there are no glaring omissions.

The figures are generally clear, informative and necessary. Some comments on Figures:
Fig. 3 – The error bars for Shannon and Simpson diversity are impossible to see on the dark blue shrub circle (this is not a major issue, given their size)
Fig. 5 The legend states “the larger the circle, the higher the correlation”. I assume “larger” should be “darker”?
Fig. S2 append legend to explain the meaning of 1,2, 3

Some minor comments:

Line 53 – change “wooden” to “woody”

Lines 105-109 it would be helpful to give authors and family of plant species at first mention in the text.

Lines 105-109 the site descriptions describe the shrub species present, but not other plant species. I might expect to see a greater difference in non-shrub species between, e.g, fen and heath, and those biotic differences might be predicted to account for variation in the arthropod community that may not be captured by shrub diversity or abiotic environmental parameters (for example, the values of “graminoid cover” may be consistent across sites (Line 292-293) but the graminoid species may differ at the species level in traits relevant to arthropods (growth form, size, palatability, etc). It would be helpful to comment on non-shrub vegetation, especially given that the supplemental data indicates all plant taxa were identified.

Line 144 is “disguisable” the word you mean here?

Lines 216, 226, etc. species authors are given for two spider species mentioned in Lines 216, 218, but not for species mentioned on lines 226, 227. Be consistent. (see also lines 228-230)

Line 227 some readers, especially coleopterists, may object to the use of the common name “ladybug”. I suggest “ladybird” or “ladybird beetle”

Lines 313-323 the first eight lines of the Conclusion are strong, but the last two sentences fade into a generic “more research is required”. Perhaps a few examples of target taxa, gradients, or sites would provide a better road map for potential next directions.

Lines 343, 367 M. De Cáceres is cited in two different formats. Check and standardize.

References – some diacritic marks are missing from author names.

Experimental design

The study is original and has a clearly articulated research question. The proposed hypothesis that arthropod assemblages differ with soil moisture and vegetation height at small scales is reasonable, but also perhaps simplistic. It would help to have some more precise predictions on the expected nature of those differences.

The sampling design is well-described, and suitable for the research question. Pitfall traps are the logical choice to collect the focal taxa (spiders and beetles). Taxonomic identity was determined at the species level, which provides necessary resolution at this scale, and voucher specimens have been deposited in a recognized institutional collection. Appropriate environmental parameters were measured.

Quantitative analyses are well-described and appropriate to the data and the questions. I am not convinced of the robustness of Indicator Species Analysis along relatively minor habitat gradients, and small spatial scales, such as these; some results may be sampling artefacts, especially with small samples sizes. But that is a minor quibble.

Line 115-117 what was the justification for different spacing (10 vs 20 m) in the fen-heath, heath-shrub plots?

Line 119-121 were traps run covered or uncovered? Both methods are used with pitfall traps in the literature so clarification would help.

Validity of the findings

The data are solid, informative, and are provided with (or linked in) the manuscript.

Line 137-138 please provide the URL and DOI for the specimen data set in GBIF. Also, I am not sure that GBIF should be credited as the author for this data set. Although they host the data, I think it would be more appropriate to cite those responsible for uploading and curating the particular data set as authors.

Lines 301-311 It is reasonable to relate spider/beetle community structure to measured vegetation parameters, but I wonder if the link made in this paragraph is perhaps too direct. The connections from overall vegetation structure to phytophagous weevils and sheet web spiders make sense, but indirect links from vegetation structure to other arthropods (e.g. prey items) to the predatory spiders and some of the beetles may warrant mention here.

---

## Round 0.2 · accepted · Accept

The authors have dealt with each comment provided by both reviewers and myself and have done so with a well-written rebuttal letter. This MS is now, in my judgement, ready for publication in PeerJ.

Thanks to both reviewers for excellent and careful reviews, and to the authors for making this a smooth process. I encourage the authors to publish the reviews alongside the MS as they add considerable value for readers.